# Unveiling the Evolution of Innovation Ecosystems: An Analysis of Triple, Quadruple, and Quintuple Helix Model Innovation Systems in European Case Studies

**Rallou Taratori** [1], **Paulina Rodriguez-Fiscal** [1], **Marie Abigail Pacho** [1], **Sesil Koutra** [1,2], **Montserrat Pareja-Eastaway** [3] and **Dimitrios Thomas** [4,*]

[1] Faculty of Engineering, University of Mons, 7000 Mons, Belgium; rallou.taratori@umons.ac.be (R.T.); paulina.rodriguezfiscal@umons.ac.be (P.R.-F.); marieabigail.pacho@umons.ac.be (M.A.P.); sesil.koutra@umons.ac.be (S.K.)

[2] Faculty of Architecture and Urban Planning, Rue d'Havre 88, University of Mons, 7000 Mons, Belgium

[3] Department of Economics, Faculty of Economics and Business, University of Barcelona, Avinguda Diagonal, 696, 08034 Barcelona, Spain; mpareja@ub.edu

[4] European Commission, Joint Research Centre (JRC), 21027 Ispra, Italy

\* Correspondence: dimitrios.thomas@ec.europa.eu; Tel.: +39-(0)-332-78-54-04

**Abstract:** Despite the rising interest in smart city initiatives worldwide, governmental theories along with the managerial perspectives of city planning are greatly lacking in the literature. It is definitely understandable that the adoption of configurational pathways toward the 'smart' 'governance' models is required as a key factor and smartness' facilitator in modern cities. In this manuscript, we display an exhaustive literature review on the importance of the n-Helix models along with a benchmarking critical approach through selected European case studies. This paper reveals the lack of exhaustive analyses for the methodological investigation, identification, and adoption of the most appropriate governance model per project including collaborative approaches. In addition, the paper deploys modular frameworks to efficiently address the continuous urban challenges, such as the rapid urbanization or the climate change.

**Keywords:** case-study analysis; citizen engagement; collaborative ecosystem; governance; innovation systems; n-Helix model; smart city

## 1. Introduction

'Smartness' increasingly appears in the literature as a promising solution for modern cities to face complex phenomena. The innovative ecosystems facilitate 'smartness' and promote citizens' engagement toward its achievement through the extensive use of information and communication technologies ([1,2]). In this work, a thorough analysis of the evolution of innovation systems is realized, unveiling their importance for collaborative synergies as a key facilitator for smart solutions.

The purpose of this work is to review the (n)-Helix models' configurations to their processual nature and how they emerge and evolve, while simultaneously presenting selected case studies and applications to provide insight (or add value) from an empirical perspective and delivered at very close timeframes (i.e., from 2014 to 2016). The time scale is a fact that allows for their comparative analysis to prove the empirical benefits of the evolution of the Helix models under the same time window investigated. The case studies analyzed in this work comprise urban projects realized in European cities with successful lessons learned about the n-Helix models' application. Finally, we aim to reveal the potential relationships between the key barriers and drivers of the n-Helix case studies, in general, and propose alternative strategic options that could ameliorate the organizational mechanisms of the innovation ecosystems' approaches.

Through a global angle, Lee et al. [1] developed an analytical scheme of the 'smartness' along with six pillars: (1) urban openness, (2) innovation, (3) partnership, (4) proactiveness, (5) integration, and (6) governance, as a lever to enable growth and city development [3]; nonetheless, this line is still under investigation [4].

The analysis of 'n-Helix' models built on 'innovative' initiatives aiming to the acceleration of the so-called 'emerging knowledge' is of particular importance. As an analytical approach, each Helix displays a distinct aspect of how societies produce, disperse, and promote 'knowledge'. As standardizing tools, they are used to instruct policy- and decision-makers in a transdisciplinary method including all the possible synergies from academia, government, and society (citizens), while both cases lead the applications along with adjoining disciplines [5]. From a more 'technical approach', the city represents a unique ecosystem for the accommodation of innovative and 'smart' systems that are perceived as 'intelligent communities', in which the synergies and the collaboration promote the social and technological originality by establishing solid engagements among the involved parties [6]. Pierce et al. [7] emphasize the importance of collaboration in smart cities' functions and the actors' synergies beyond traditional processes.

Literally, the term 'ecosystem' is usually associated with 'smartness' [8] as a concept, which reflects the 'information', 'communication', 'collaboration', and 'technology', but it resembles varied kinds of other components, as proposed by Benson [9] (Figure 1), which require more universal approaches on how ecosystems bring together technology, government, and society to achieve 'smart' objectives of the city planning [10].

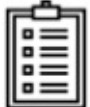
**Government Efficiency**
Governance effectiveness and ease of doing business

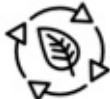
**Sustainability**
Environmental, energy, water and air quality management

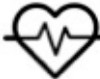
**Health and Wellness**
Mental, physical and social care and well being

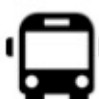
**Mobility**
Transportation, transit and traffic management

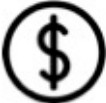
**Economic Development**
Business, employment and productivity

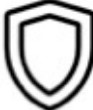
**Public Safety**
Welfare and protection from crime, hazards, and disasters

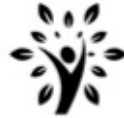
**Quality of Life**
Standard of living, satisfaction and happiness

Source: StrategyofThings.io

**Figure 1.** Smart city 'ecosystems' [10].

From the other side, Ooms et al. [3] highlight the importance of 'governance' to stimulate 'smartness'; thus, a city is viewed as an ecosystem composed of physical boundaries (location, geography, and topography), its population, and the interactive flow (i.e., functions, society, growth, etc.) [11].

To the roadmap of 'smartness', cities seek more innovative standards of production along with technological achievements in their organizational dynamics [12]. It appears that 'actors' collaboration and synergies remain the primordial challenges for the 'smart' success, whereas the degree of accomplishment is a factor of technological, governmental, institutional, and beyond integration, [13], without neglecting the importance of participatory processes and the citizen engagement to the 'smart action plan' [14].

Nonetheless, the 'ecosystem' has a wide spectrum of interpretations varying from its technical to the more human-oriented and social approaches toward a 'smart' urban transformation [15] aiming at the benefits of the QoL and the long-term and 'intelligent' city development [16]. To better visualize the design and implementation of 'smartness' through more tangible results, Appio et al. [6] organized Giffinger's typical 'smart' model of the six pillars following the model of Hutchinson's pyramid (Figure 2):

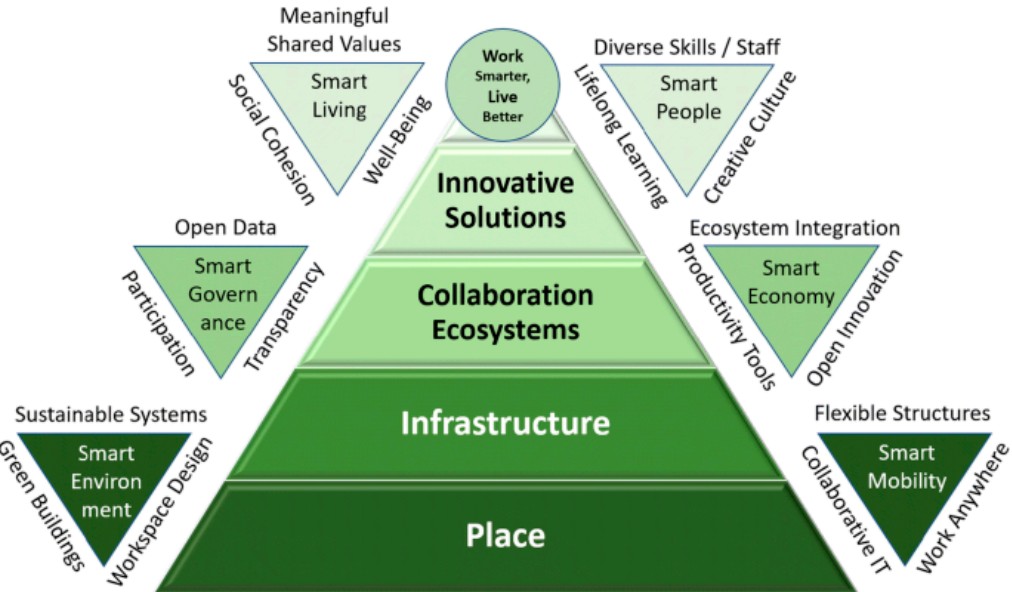

**Figure 2.** An adaptation of Hutchison's pyramid proposed by Giffinger for the understanding of the 'collaboration city ecosystems' [6].

Nam and Pardo [17] relate the 'smartness' in cities to human-oriented and technological aspects and consider it as a lever for innovation and knowledge. At the same time, Dameri [18] enforce the 'smart' idea to accelerate the economic growth on cities but also to reply to the citizens' needs. Complementary to these arguments, Van den Bergh et al. [19] urge the concept of 'ecosystem' to deal with the urban modern challenges, which involve numerous and conflicting actors. Abella et al. [20] consider the concept as an added value to promote the 'digital' integration and the information processes.

From the other side, Ooms et al. [3] conceptualize the term 'ecosystem' along with: (1) the 'smart' attributes, including the different stakeholders, (2) the governance, and (3) the necessity of being 'sustainable' in long-term horizons. Chan [21] describes the SSC as *'an ecosystem comprised of people, organizations, policies and processes towards desired outcomes—a city adaptive to its environment and responsive to technological evolution to accelerate and to facilitate its actions'*, in which sustainability is prioritized. Figure 3 represents the 'smart city' model in 'layers' organization, in which each has a particular role for the accelerating integration of the technology to mutate the ecosystem. This process is usually supported by the synergies between stakeholders, while a key factor for its success is undoubtedly the 'governance'.

Schaffers et al. [10] highlight the significance of collaborations and synergies in ecosystems for the 'smart' sharing of resources and the experimentation of technologies and applications for exploring the city challenge. Three pioneering perspectives are argued for the smart city ecosystem (Table 1), which are summarized on:

- The Internet and research, focusing on a technological-oriented contribution to the smart city ecosystem;
- The policies around the city planning and development;
- The exploration of innovative solutions stimulating the participatory processes toward 'smart' solutions.

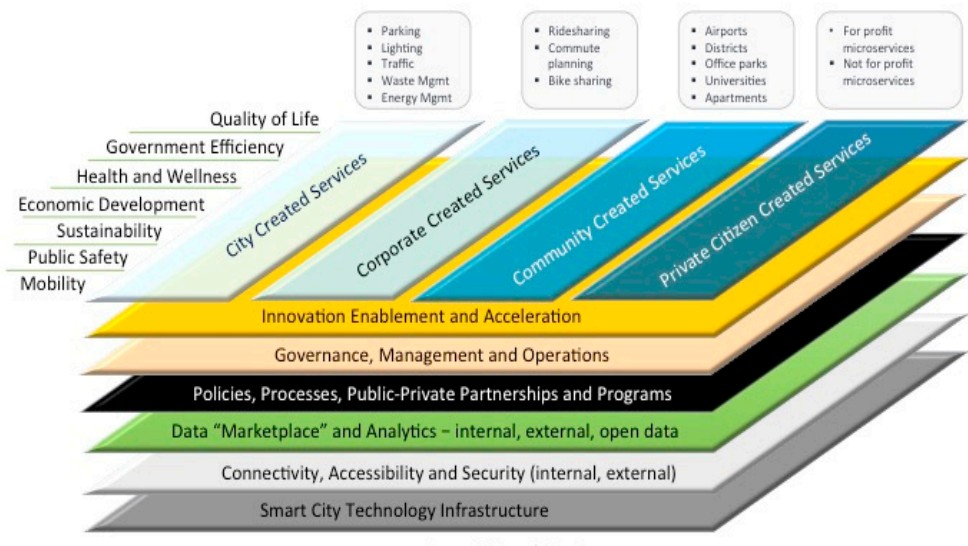

**Figure 3.** The 'smart city' model in layers [21].

**Table 1.** Perspectives of synergies developed in smart city ecosystems.

|  | Internet and Research | Urban Development | Innovation |
|---|---|---|---|
| Stakeholders | Scholars<br>ICT companies<br>City and EU actors | City policy actors<br>Citizen platforms<br>Associations | Living lab managers<br>Citizens Government<br>Research institutions |
| Priorities | Technical and other challenges | Infrastructure and other services | User-driven open innovation<br>Citizen engagement |
| Resources | Experimental facilities and pilot projects | Development plans | Methodologies and tools |
| Policies | Experimental research | Policies to stimulate innovation, business, and procurement | Innovation projects, open and collaborative sharing data |

The study is structured accordingly: after the introduction of the research motivation, it emphasizes the contextualization of the topic. The Section 2 focuses on the 'governance' as a driving force to enable motivations and action plans to the city planning and the notion of 'ecosystem'. Section 3 analyzes the theoretical approaches of the selected n-Helix models, highlighting the benefits through practical applications on European case studies. Section 4 emphasizes the benchmarking results, while to this end, Section 5 discusses the main findings of this study and the perspectives for future work.

## 2. Collaborative and Smart Ecosystems. The Role of 'Governance'

The stepping stone for the emergence of the 'smart city' movement was another concept that made its appearance in the last decade of the 20th century, the so-called 'smart growth' [22]. It supports innovation in policies for urban planning, while having been widely adopted at the industry level after 2005, as it supports the deployment of complex information systems toward the inclusion of services and infrastructure. Hitherto, the concept has been gradually extending and updating to its current conceptualization state, which practically encompasses any form of technologically led innovation exploited for the planning and development of cities. Pierce et al. [7] make a distinction between the terms 'sustainable city movement' and 'smart idea', even when they admit the existence of a connection between the two. Specifically, they discuss that the main principle of the latter is the exploitation of technologically driven innovation for sustainable transitions, which is relatable with a framework that advocates strong interlinks among stakeholders and



ecosystems. The existent literature body revealed three main categories of ideal/typical definitions of smart cities: as cities exploiting smart technologies for their smart transitions, as cities that focus on human-oriented innovations for their smart transitions, and lastly, as cities that perpetually strive to update their governance in a more ecosystemic approach [23]. Some elements that have assisted in the wider recognition of the smart city concept's value are the potential: amelioration of citizens' QoL, capabilities it creates for achieving sustainability (i.e., environment, economy, and society), and increase in cities' attractiveness [24].

Toppeta [25] in his definition of the 'smart city' focuses on the combination of technological and ICT capabilities with designing, planning, and organizational approaches that aim to accelerate the administrative and bureaucratic processes in order to formulate innovative solutions that facilitate the management of complex city mechanisms. Caragliu et al. [26] expanded the aforementioned orientation by complementarily integrating the concept of 'participatory governance'; in particular, they declared, *'we believe a city to be smart when investments in human and social capital and traditional and modern communication infrastructure fuel sustainable economic growth and a high quality of life, with a wise management of natural resources, through participatory governance'*.

To visualize the discussed definition, a scheme connects the smart cities' envisaged smart visions with the exploitation of technology and ICT through integration, the human factors to be taken into consideration via learning, and the institutional factors that could be reinforced via participatory governance, which is shown in Figure 4 [27]:

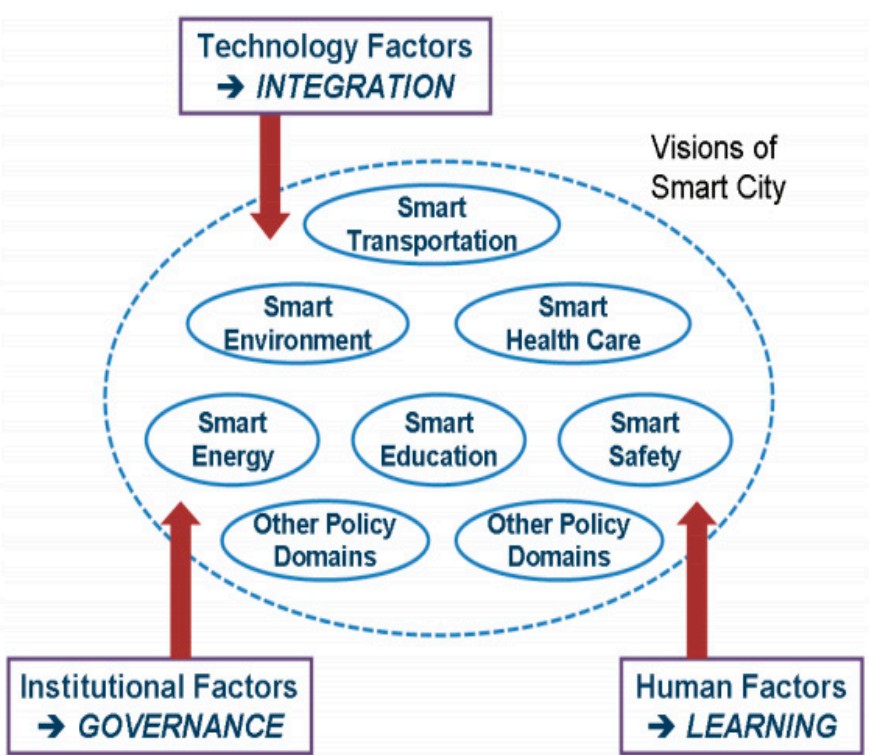

**Figure 4.** Visions of 'smart city' [27].

Governance encloses all governing processes, while the governing authorities could be a government, market or network, formal or informal organization or territory, and the landing could be delivered via laws, norms, power, or even language [28].

Tomor et al. [29] propose three governance components that entail the concept of 'smartness' cities:

- Governmental organization: This yields for the local stakeholders' commitment toward the development and deployment of 'smart' initiatives with the use of ICT and their appropriately designed operational management;

- Citizen participation: this ensures the citizens' engagement and empowerment in policy decision-making processes and through different stages of the realization of the project [30];
- Use of technology: Various digital technologies could be exploited to maximize the impact of participatory processes (e.g., communication and management platforms, project dissemination websites, discussion forums, and meetings).

Along the same lines, Ooms et al. [3] display the need for the emergence of governance models with more complex structures and processes, whilst in the pursuit of 'smartness' in cities through the active inclusion of multiple actors. The literature review revealed two different focus orientations while exploring the significance of 'governance' in smart cities: the roles of specific urban actors and stakeholders [31], or the role of human capital. Nonetheless, the exact causal relationship(s) between 'smartness' and the role of 'governance' are yet to be fully understood and defined.

Deakin [32] mentions that, under the policy lens, the 'governance' is linked with participatory processes via academic leadership and corporate strategies. More specifically: *'the capacity to process the transition reflexively from creative to intelligent and as part of the cities' "smartness".'* This notion strengthens the collaboration between all involved stakeholders within 'smart city ecosystems', which subsequently leads to successfully reaching the envisaged objectives and city 'innovation'. Schaffers et al. [10] highlight the significance of synergies and partnership in sharing research and resources to create cooperation models. Participatory governance and citizen involvement are, in any case, key components in the 'smart city' framework and are at the core of 'smart' initiatives. These two keep decision making active and ensure the cooperation of multistakeholders throughout a project [10].

During SINFONIA EU project, researchers [33] conducted a feasibility study and analyzed the experiences of over a hundred completed and ongoing smart city projects. They identified public participation, cooperation between different stakeholders, and the long-term political commitment to be the most powerful drivers. Additionally, they revealed that public participation is furthermore the most utilized factor for overcoming the key recognized barriers. Thus, special attention is given in the citizen participation parameter in the selection of the case studies and their respective conclusions.

## 3. Analytical Framework of Helix Models and Case-Study Applications

In this section, an analytical description of the studied Helix models is provided, while particular cases explain their applications in reality.

From a structural view, the creation of 'collaboration' in cities promotes the innovative synergies and the citizen awareness and engagement of public and private involved institutions. Thus, the 'smart ecosystem' involves a wide spectrum of stakeholders engaged in a continuous collaboration on a human-oriented knowledge and learning process toward 'intelligent' solutions to urban challenges. The Triple Helix Model was firstly proposed by two scholars: Etzkowitz and Leydesdorff [34] and became the pioneer of 'smart' collaborative ecosystems. The traditional stakeholders in charge of developing and sharing innovation in the industrial sector sphere and creating knowledge in the academia interact with the political dimension of transferring the knowledge to promote the economic growth in cities through top-down approaches. This model stresses the importance of different pillars, commonly known as 'helices', to generate 'innovation' in the precedent sectors: academia/industry/government, placing an emphasis on the 'tri-lateral' interconnections contextualized by the broader dimensions developed on forthcoming evolved forms, such as the Quadruple or the Quintuple Helixes.

Appio et al. [6] emphasize the benefits of the n-Helix models (and more precisely of the 'Triple') for the economic growth and attractiveness and the environmental challenges, while the importance of the evolved models of 'Quadruple ' or 'Quintuple' approaches accentuate the significance of a continuous and interactive process involving diverse stakeholders and investors, without neglecting citizens. Figure 5 [6] illustrates the evolution of the n-Helix models in regard to the knowledge production and innovation.

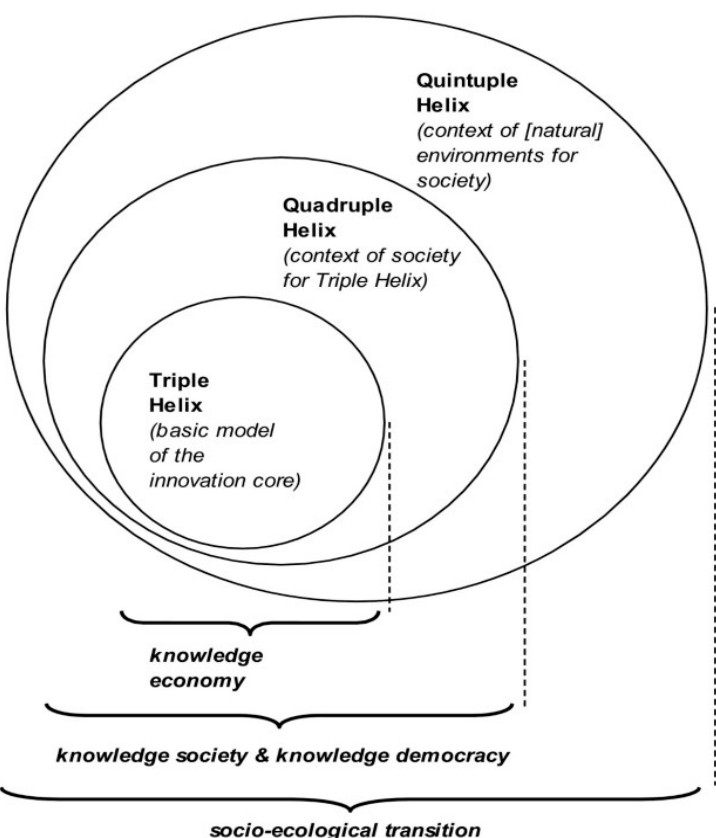

**Figure 5.** Knowledge production and innovation models [6].

In fact, all the (n)-Helix models share attributes of complexity in their functions [35]. Firstly, the models are inter- and trans-disciplinary, meaning they come from both natural and social sciences, highlighting what is 'innovative' as well as the importance of multiple subsystems (of actors). At the same time, their dynamic changes forge societal, institutional, and organizational contexts, which enable (or hinder) the innovation processes by expanding it beyond the pure technological aspects [36].

McAdam and Debackere [37] claim that the Helix models have the ability to understand the rising interactions between the developed subsystems as a primordial attribute of the innovation processes and beyond this to assess the interactions between the components of their spectrum. In this context, the Triple Helix Model has been particularly influential in the literature, while as a continuity the Quadruple and Quintuple not only promoted the policy making but also encouraged the transdisciplinary analysis of the sustainable development. This link is thoroughly explained by Etzkowitz and Zhou [38], who stress the connection in particular with the SDG9 regarding the infrastructure, the industrialization, and the innovation. A more detailed presentation of the three representative Helix models is followed along with selected case studies in Europe.

### 3.1. Triple Helix Model

The central idea of this model is that the 'ecosystems of innovation' in cities are developed along with three types of agents [39]:

- Universities: which have the main behavior as a 'magnet' to stimulate the scientific and technological knowledge;
- Industrial sector: which is the key to boost the creation of economic growth;
- Government (local, regional, national, and international): which has an active role of actions, management, and land-use policies.

The Triple Helix Model is one of the most referenced in the literature used to define an innovation ecosystem and postulates that the interaction of the three elements, as

mentioned above, to improve the requirements for 'being innovative' in a knowledge-based society unveils the need for interactions among its elements, as illustrated in Figure 6 [39]:

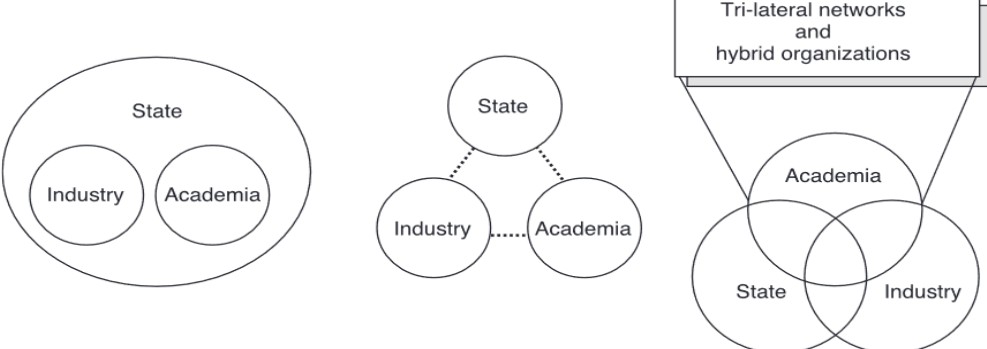

**Figure 6.** From 'etatistic' and 'laisser-faire' to the model of 'Triple Helix' [39].

Appio et al. [6] argue that in reality in cases in which the model was developed, the environment has been more competitive and 'economically attractive' enabling the pillar of 'smart economy'. Nonetheless, the model does not work satisfactorily in all its applications due to the influence of barriers between the different parties involved and the lack of interfaces used to create limitations on its use ([40,41]).

According to Bellgardt et al. [42], the Triple Helix approach cannot be usefully applied to the actors' interactions in residential development in science cities, since is not suitable for mediating the process of planning for this urban approach. Firstly, in terms for urban planning development, demand for housing is not distributed across the three helices in an even way. The Triple Helix Model undertakes that the helices interact at the same time under the fact that this interaction creates added value; industrial stakeholders choose not to engage in the planning process because they predict no immediate economic benefit [43].

Case-Study Analysis: Berlin-Adlerfshof Science City

An example of the Triple Helix application is studied in the project of Berlin-Adlershof Science city as an idea of integrating three complementary elements: university research, institute research, and private businesses, to serve as keystones of a dynamic cluster with a focus in technology development, with an initial support from the government side toward the settlement of research institutions reinforced by new companies [44]. This particular case was selected in order to show the weaknesses and threats that may hinder a successful Triple Helix implementation and contemplate the final importance of them and the absence of public participation to the quality assurance of the project. Berlin-Adlershof is the largest cluster dedicated to high technology in Germany, integrating industrial, media, and scientific facilities on a site [42], while by 2016, the park hosted more than 1000 companies and scientific institutions that attracted 15,996 people for work and 6524 for study [45]. Science and Technology Park Berlin Adlershof is one of the most successful high-technology places in Germany and among the 15 largest industrial parks in the world (Figure 7) [46].

In this case study, following the Triple Helix model, the stakeholders are represented by all commercial enterprises (industry), university and research institutes (academia), and the Senate Department for Urban Development and the Environment (SenStadtUm) and the administrative district (Bezirk) of Berlin Treptow-Köpenick (government). The government of Berlin defined an entity, called WISTA, as a public and private partnership company, which is responsible for operating the park and the construction, lease, and operation of the incubator and technology centers [47].

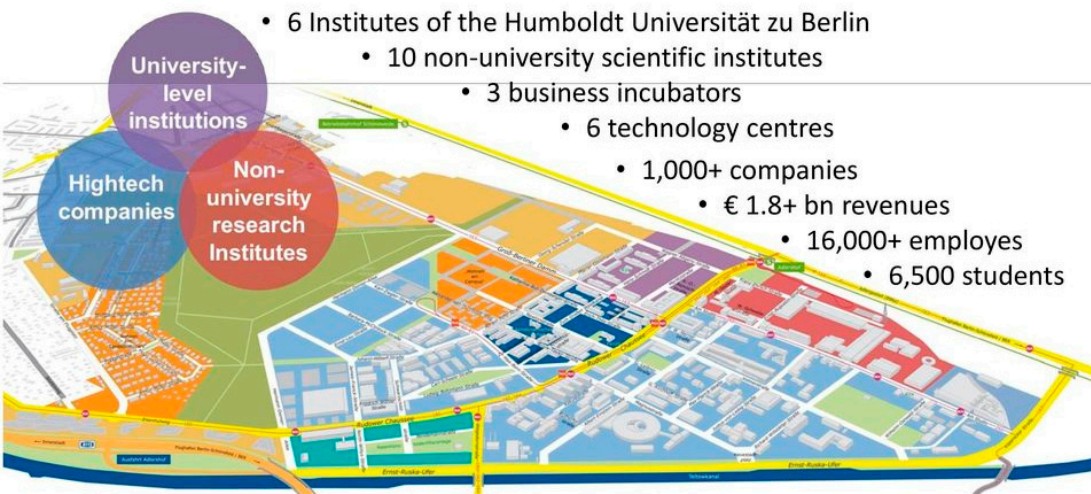

**Figure 7.** Illustration of the project of Berlin-Adlershof Science City [46].

The Triple Helix Model was considered for analyzing whether the housing planning for the innovation hub can be integrated and envisioned as a pioneering product in terms of the collaboration model. The Triple Helix approach is exploited to enable the organization and project management as well as the synergies among its involved parties (stakeholders), which are related to different processes of innovation. Bellgardt et al. [42] focused on a study on the sociocultural aspect in the urban development context, regarding science cities, since this sociocultural approach was discarded in previous research on the Triple Helix Model (Figure 8) [42]:

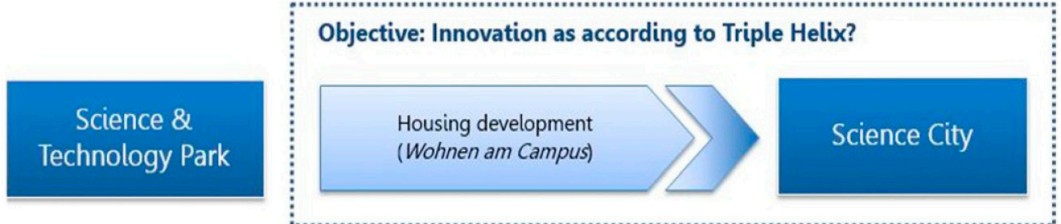

**Figure 8.** From an innovation park to a science city (project of Berlin-Adlershof [42].

There are several success factors in Berlin Adlershof case. Among them, the government provided support since the beginning, in terms of the financial approach, which gave an encouraging ambience for innovation and creativity. In addition, this governmental actor supported effective park management and development through the public private partnership and the politically planned determination [47]. Additionally, an important factor of success is the existence of a long-term plan to promote and expand the site and create links and working relations between the three Helixes.

On the other side, citing the negative factors for this model, the stakeholders' interactions are reliant on their resources. The capability for the academia (Humboldt University) to participate was limited by its financial and recruitment resources, in comparison with the industrial actors. Lastly, as there is no existence of an external organization, a fusion organization from the three helices intersect does not function independently.

### 3.2. Quadruple Helix Model

As explained previously, the Triple Helix Model is a scheme for action of knowledge-based economy [48]. In this evolution, the Helix encounters the 'citizens' in a Helix, recognizing their increasing role of the society [49] and, as a further Helix, the economy associated by the 'media-based and culture-based public'. Inspired by Carayannis et al. [49], the fourth Helix reflects on phenomena such as the 'media-based democracy' by being

at the same time human-centered and in favor of knowledge. The main objectives of the Quadruple Helix Model are summarized by:

The lack of cohesion and presence of a limited intellectual exchange in the shaping of the smart city modelling (Figure 9):

- The need to offer a 'technological determinism', in which utopia is imposed within the need for innovation to reply to the requirements of urban sustainability;
- The legitimization of research trajectories among the relevant publics aimed at having public impact on more sustainable solutions [50].

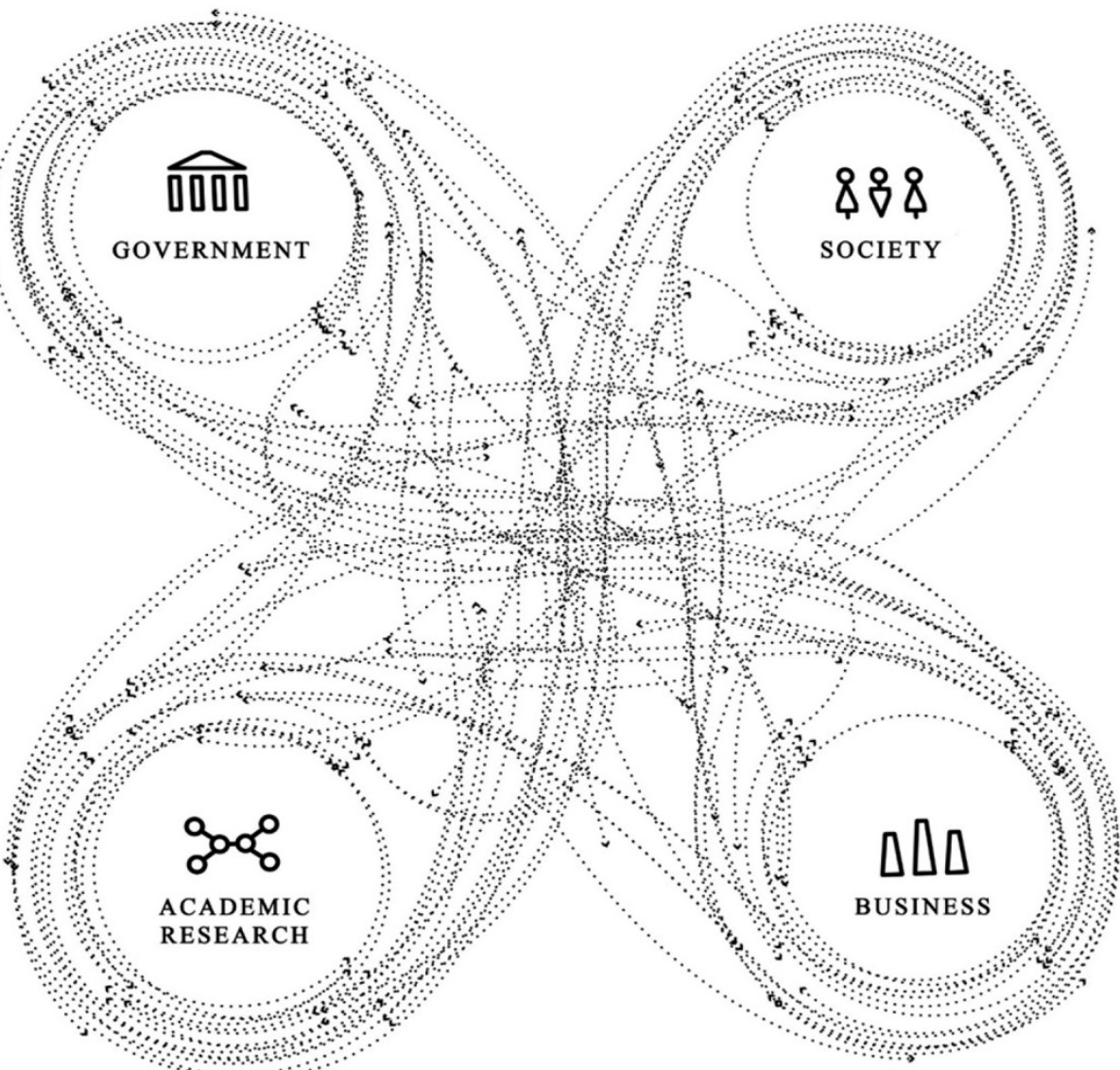

**Figure 9.** Representation of the Quadruple Helix Model [50].

'The Quadruple Helix Model introduces the focus on the pillar of culture and the importance of constructing and communicating the 'public awareness' as this relation has an impact for every dimension of the system' [50]. Nordberg [51] defines the fourth Helix as the more 'cultural' dimension and the backdrop toward the roadmap to innovation, while Ivanova [52] discusses this topic from a more systemic view, focusing on services, arguing that the Quadruple Helix Model not only addresses the consumer but also the communication and the media. From the other side, Höglund and Linton [53] argue that the fourth Helix is not a separated additional Helix but an integrated part of the society and its significance is to reply to the citizens' requirements. Nonetheless, despite the undeniable

contribution of the Quadruple Helix model, there is a methodological challenge on the way of the citizens introduce their public perspective and also how the different actors define their functional role of the society as a fourth pillar and in collaboration with the innovative processes.

Case-Study Analysis: The Flottsund Bridge

An empirical basis that explores how the Quadruple Helix Model unfolds and functions is the renovation of a bridge located in Uppsala, whose local authorities initially considered integrating a renewable energy technology by collaborating with researchers involved in academic entrepreneurship [54]. This project was originally designed to follow the Triple Helix Model, but the necessity to include the citizens later on in the project makes it a good case to stress the significance of thorough planning that engages the public at the early stages of decision making. Failure to do so may lead to catastrophic outcomes and may, therefore, undermine the whole scope of a project.

This project was initiated in 2014 by the Regional Development Office of Uppsala (hereinafter referred to as the "regional office"), whose original leader had a personal interest in environmental issues. Situated in Sunnersta (Southern Uppsala), the Flottsund bridge is a key connection for the city and the county and is a vital route for vehicular, bicycle, and pedestrian traffic (Figure 10) [55].

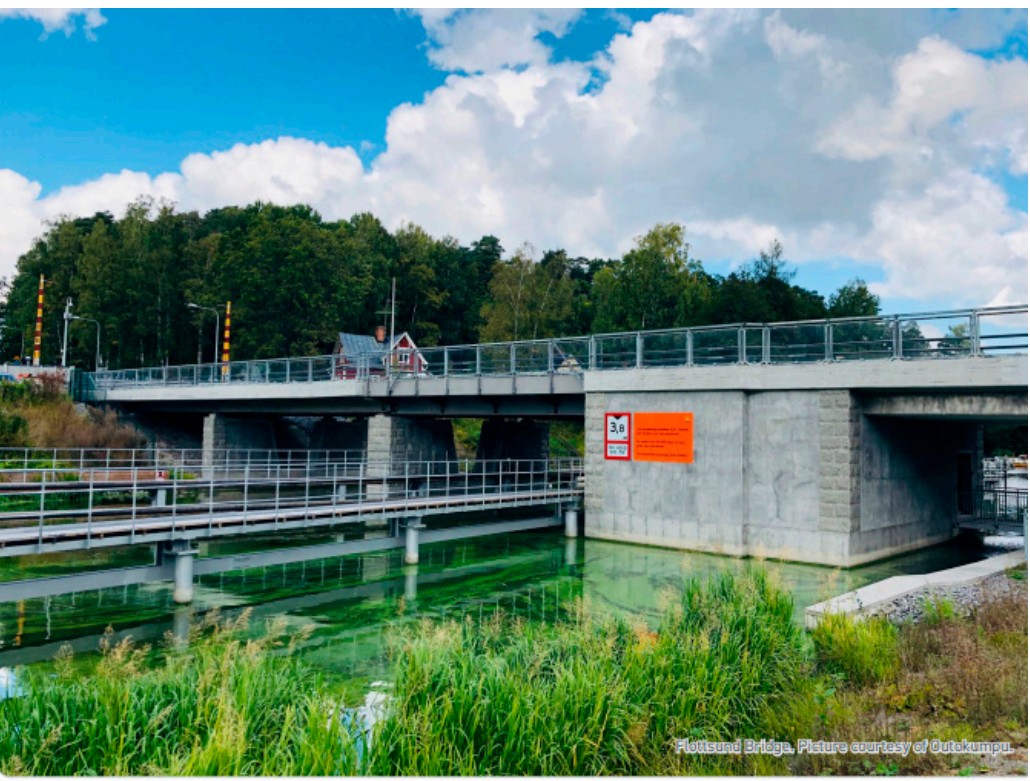

**Figure 10.** The Flottsund Bridge [55].

The late engagement and inclusion of the citizens in the Flottsund bridge renovation project, however, proved ineffective as citizens already felt left out of the decision-making process; open meetings meant to create a dialog with citizens ended up as being one-way conversations, and local authorities from the regional office only engaged with the residents to seek approval for a project that had been already decided upon without the latter's prior involvement. Since the concern of the citizens primarily lay in the disruption that would be caused by the renovation, the regional office marketed the project as one whose social value was an upgraded recreational area for the community. In the end, the renovation

project proceeded without the integration of the marine power current turbine, thus failing to achieve the original objective of commercializing said technology.

### 3.3. Quintuple Helix Model

Carayiannis et al. [56] explain the Quintuple Helix Model as an interdisciplinary and transdisciplinary at the same time in the sense that the fifth Helix structures a more analytical aspect for a dynamic involvement of all the involved parties. The most important ingredient of this evolved model, apart from the active and more 'human-oriented' approach, is the resource of 'producing' knowledge through a circular process between the subsystems (society, economy, etc.). Thereby, the Quintuple Helix Model visualizes the importance of collectivity and the exchange along with the education, the economy, the environment, the society, and the political systems as represented in Figure 11 below [56]:

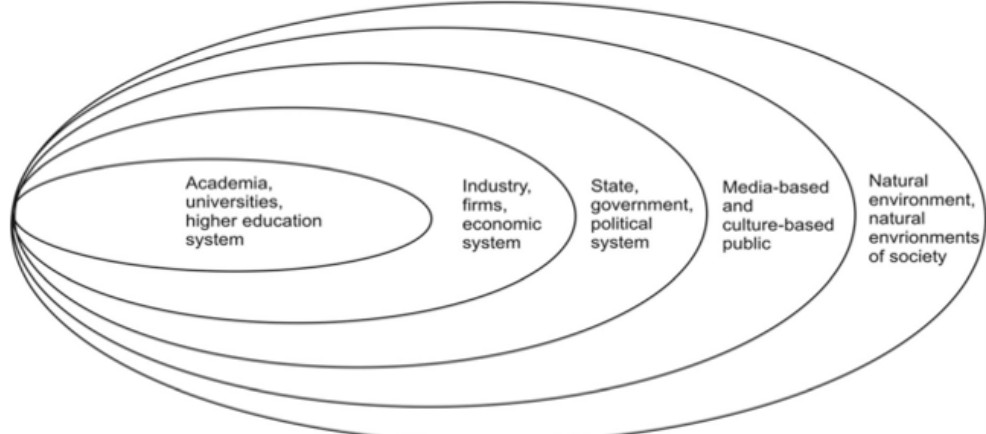

**Figure 11.** The subsystems of the Quintuple Helix Model [56].

The Quintuple Helix Model of innovation adds a fifth dimension to the innovation processes, with the emphasis on the nature and social ecology [56]. Nonetheless, a question, which usually remains unsolved within the Quintuple Helix model, is the way to connect the five helices in an innovation process. To this topic, Markard et al. [57] argue that a response is around the 'ecology', which is the interdisciplinary either of relations between living organisms (social) or either between them and their environments (natural), which is subsumed as an 'ecosystem'. All these concepts around the Quintuple Helix Model converge in the society–nature transition. Thus, in the Quintuple Helix, the focus is around the translation of the environmental and ecological issues by identifying them as 'drivers' for future knowledge and innovation.

Case-Study Analysis: The 'Remote Control of Ventspils Water Supplying System'

The examined case study under title "Remote Control of Ventspils Water Supplying System and Smart Metering Services", was completed in 2016, while the main leader of the project was the Municipality of Ventspils. The project's main objective was the creation of a remote control and smart metering system for a water supplying system in Ventspils city and the social sensibilization, education, and awareness of the services' use, as well as the improvement of the synergies among the users and the involved investors or other stakeholders. The literature review, while trying to select a Quintuple Helix case study, revealed that the maturity of this concept yields the realization of highly sustainable and resilient projects that share multiple common characteristics in the governance orchestration mechanisms; this is a typical early application example of the examined model found in the literature. The project has a variety of tasks, which were attributed to the different involved stakeholders as briefly presented below [58].

Ventspils University College and Kaunas Technological University were responsible for the creation of a novel electronic equipment for reading and transmitting data from wa-

ter network nodes, appropriate software to respectively store and process all the retrieved data, and finally e-services that exploit the gathered data and software. Entrepreneurs such as the small enterprise, Smart Meter Ltd. and Ventspils UDEKA (municipal enterprise of water supply), were responsible for equipment testing, services implementation, and general systems' maintenance after the project's end.

The nongovernmental professional institution Latvian Internet Association's (LIA's) primary goal is to educate society in internet technologies' usage and to ameliorate SMEs' business environment in the country, acting as the moderator between government and entrepreneurs. Therefore, they were eager to participate in such a project, due to its significant potential for positive influence on the development of e-services within Latvia, and contributions toward society's education and entrepreneurs' engagement. Their role was to create education materials for the public and that were relevant to the upcoming e-services, while additionally assisting the municipality in the organization of seminars to introduce citizens in the new e-services possibilities. Finally, they acted as one of the dissemination stakeholders, along with the media.

Taking into consideration that proper dissemination plays a vital role in the success of Quintuple Helix innovation models, as well as that the local media, can contribute toward this objective by providing proper communication with the local community, and the participation of the local newspaper named "VentasBalss" was secured. Moreover, other typical social media channels were exploited for the cause of the project by all its partners.

The project belongs to the Quintuple Helix Model innovation ecosystem category, as it successfully achieved the cooperation of various stakeholders from the government, academia, industry (with SMEs and mass media engagement), and with an environmental-oriented goal (e-service for a remote control and smart metering system for a water supplying system) and socioecological outcomes. Moreover, special attention was given to the education of society in ways to exploit the new electronic services and to improve the levels of cooperation between the residents and the municipal utility companies, thus accumulating greater experience and knowledge in cooperation within complex innovation ecosystems. The base of cooperation amongst the different involved stakeholders was established with the development and usage of a communication and management platform for the project.

One of the key factors that secured the success of the project was the careful selection of the stakeholders to be engaged and the roles that they were assigned, something that accelerated the progress of the project, ensuring top quality and success.

Specifically [59]:

- For projects that target local goals, it is wisest to involve the respective local government. In this case, the water control and smart metering system was targeting the sustainable transition of the whole Municipality of Ventspils city, and thus, the respective municipality was the main contractor of the project. Via the Quintuple Helix cooperation relationships established between this and the rest stakeholders, the outcomes achieved were mainly two-fold: on the one hand, the municipality updated and modernized their structures and services, with the realization of the novel remote-control water supply system for end users and utilities, while on the other hand, the communication with society was greatly facilitated and strengthened, as the municipality ensured transparency throughout the project's processes for the society and additionally offered organized educational approaches, via both the existing municipal portal and the local mass media, for the introduction of the new e-services to the local community;
- Involving universities in social–ecology projects increase the potential to achieve high-quality technological achievements. Most importantly, however, it allows for scientists, scholars, young researchers, and students to engage in the philosophy of technological progress aligned with social benefits for the society;
- NGOs also play a key role, as they have the potential to facilitate the communication amongst industry and the society, contribute toward the expansion of e-services do-

main, and accumulate new knowledge regarding the communication and cooperation with new partners, for example, academia;

- Entrepreneurs (in this case, SMEs) are benefited and contribute both toward the exploitation of novel equipment and services and from the knowledge they can gather on the information and development workflow management and the communication practices with society and potential audiences;

- The inclusion of the (local) media is also important for the success of the project, as it has the potential to internally monitor development processes. The simplest way to practically introduce a local public audience to new e-services is via points of easy access for them, such as the local media, municipal portals, and newspapers.

Another identified vital point for the success of the examined project was the proper and thorough information of the local community and the increase in their self-awareness, as mentioned in the initial goals of the project ' . . . *and to educate society in how to use the new electronic services, and to improve levels of cooperation between the residents and the municipal utility companies'*. This was achieved through the cooperation of all stakeholders, while the Latvian Internet Association prepared the content, and the dissemination was performed via the mass media and municipal portals. Efficiently including and educating society in Quintuple Helix Model projects is pivotal, as through their increased self-confidence on social ecology, social, and smart economy (i.e., reduction of bills for water and a fairer price for consumption with simultaneous understanding of the importance of achieving overall sustainable development), they facilitate a holistic sustainable development. With the services of this project, citizens were able to follow water usage and to engage in the nature recourse saving program. Moreover, in projects such as the one under investigation, where e-services are created for the environmental monitor and management from end users, the success of the results might be measured in statistical savings, yet they are explained via the citizens participation, something not quantitively discussed or measured in the case-study literature. However, because of the proper planning from the involved project partners for making the information easily available and providing efficient training on the ways of the new e-services usage, the citizens started to exploit the e-services, and this is what contributed toward the successful water-management-quantified results.

Finally, another recognized strength of this project was the establishment of an effective information exchange, cooperation, and communication channel among stakeholders, via the development of a management and communication platform. This allowed for the emergence of a cooperation model between project partners from various groups (i.e., municipality, academia, nongovernment, SMEs, and mass media) and to the adequate management of resources (e.g., technical, human, economic, time, etc.) for the final completion of a successful project with the new products and services and within the originally set time horizon.

### 4. Results and Discussion

A benchmarking (SWOT) analysis of the main findings of the case-study analysis is explained analytically in Table 2.

Based on the aforementioned key positive and negative factors identified, the four resulting areas of the SWOT analysis are set against each other to understand and reply to the queries of:

- Which opportunities can enforce specific strengths, or which strengths can be used to exploit specific opportunities?
- Which opportunities can contribute toward the minimization of the weaknesses?
- How can strengths be exploited for the minimization of threats?
- How can weaknesses be minimized to eliminate threats?

Table 2. Benchmarking (SWOT) case analysis of selected applications.

| | Strengths | Weaknesses |
|---|---|---|
| Innovation Park to a Science City | Interactions of actors<br>Regular evaluations and auditing processes<br>Existence of public-private engagement model<br>Existence of expertise, awareness, and methods for designing implementation of new technologies and solutions | Limited cooperation and interaction between the three helices in planning steps<br>Uneven distribution of demand limits the interactions<br>Model basically assumes the entrepreneurial model<br>Lack of adequate communication and transparency between project participants and the public to raise awareness<br>Inadequate/general definition or documentation of processes |
| Flottsund Bridge | Efficient and constraint-driven project management<br>Public participation<br>Existence of expertise, awareness, and methods for designing and implementation of new technologies and solutions<br>Interoperability between systems | Short-sighted planning<br>Lack of adequate communication and transparency between project participants and the public to increase awareness<br>Exclusion of citizens from the earlier stages of planning and decision-making processes<br>Lack of cooperation and distrust between different stakeholders Inertia<br>Lack of binding agreement among involved parties<br>Lack of integrity from the regional institutions<br>Failure to integrate varying stakeholder interests into a shared vision<br>Absence of well-defined or documented-in-detail processes |
| | Opportunities | Threats |
| Innovation Park to a Science City | Long-term strategies<br>Strong political commitment over the long term<br>Foster of innovation processes | Insufficient financial support<br>Absence of financial models suitable for the innovation to address stakeholders<br>No attention to values of citizen engagement and management<br>Merged organization from the three helices intersect was not functioning independently<br>Potential conflict of stakeholders' interests in collaboration ecosystems |
| Flottsund Bridge | Enabling political framework for renewable innovation and entrepreneurship<br>Potential testing and upscaling of the project<br>Additional income generation from green tourism and commercialization<br>National roadmaps, strategies, and policies for energy goals<br>Existence of affordable and mature technologies suitable for local conditions | Public acceptance of technologies<br>Insufficient financial support<br>Changes in the administration and political agenda and lack of leadership<br>Potential conflict of stakeholders' interests in collaboration ecosystems |

The key findings after comparing the case studies, in relation to the most powerful drivers as mentioned in a previous section (i.e., public participation, cooperation between different stakeholders, and long-term political commitment) are summarized below. The causal relationships between the examined drivers and barriers are unveiled, and robust organizational mechanisms and strategies for achieving efficient innovation 'ecosystems' are proposed [59]:

(a)  Connection between strengths and weaknesses

Strengths: Public participation and proper and thorough communication between project participants and the public to increase awareness and involvement

Weakness: Inertia

Typically, it is expected that a behavioral change does not happen easily, which is something that applies to matters of implementation of new technologies and services. However, this weakness could be eliminated by the realization of relevant advice campaigns for the public. It is highly recommended that the dissemination of knowledge and information must be 'translated' and handed over in a simplistic way for everyone to understand. These advice campaigns have the potential to lead to easier end-users' behavior changes, as they have advice for their smooth transition into the new services or technologies. Moreover, by actively involving the educated citizens into the decision making, they are more likely to regain their trust in the proposed project and make conscious choices and not intuitive ones.

(b)  Connections between strengths and threats

Strength: Public participation and proper and thorough communication between project participants and the public to increase awareness and involvement

Threat: Public acceptance of technologies

The lack of public acceptance for new technologies might be traced back to the lack of appropriate information on costs and benefits of the technologies, which subsequently leads to the lack of trust that the upcoming decisions are beneficiary for all involved stakeholders [33]. By actively engaging the public into discussions with the rest stakeholders, the knowledge of the former on costs and benefits increases, and thus misconceptions are expected to diminish. Along the same lines, by comprehensively educating (i.e., advice or educational campaigns) the public on the new technologies and transparently involving them into the decision making on technological activities, the feeling of suspiciousness is expected to decrease significantly.

Strength: Existence of expertise, awareness, and methods for designing and implementation of new technologies and solutions

Threat: Potential conflict of stakeholders' interests in collaboration ecosystems

Even though different projects may apply similar smart governance solutions during the realization of smart city projects, it is worth mentioning again that each project refers to a different city with different contexts and different ecosystem balances. This means that the smart governance mechanisms cannot be generalized for other cities or even projects within the same city, as the balances among the involved stakeholders are liquid and subject to change per case. However, this issue could be addressed by the integration of expertise, awareness, and methods for designing and implementing projects. Multidisciplinary teams of experts could analyze the social, economic, political, environmental, and legal conditions of the project and design and orchestrate localized governance mechanisms, which would respond to the specific needs of the project.

(c)  Connections between opportunities and weaknesses

Opportunities: Political commitment over the long term and national roadmaps, strategies, and policies for energy goals

Weakness: Lack of adequate communication and transparency between project participants and the public to increase awareness

There are cases that the communication and trust between project partners and the public are insufficient. Nonetheless, it is expected that the interest of the public and their efforts for communication with the other partners increase, if they are ensured that there will be stability and responsible actions toward the implementation of a project, independent of the political party in charge. This feeling of collectively striving for a common sustainable goal could be magnified by the existence of national roadmaps, strategies, and energy for energy goals on the long run.

(d)  Connections between opportunities and threats

Opportunity: General environmental issues and (socio)-ecological transitions

Threat: Public acceptance of technologies

A potential absence of acceptance for new technologies and services may lead to delays regarding the progress of the project. However, the treatment of environment and the adoption of a 'social–ecology' mentality could alleviate this threat. Empirically, it is known that the treatment of environmental issues draws investors' interest and leads to both increased public participation and acceptance of smart city projects [33].

These are the key identified ways that innovation ecosystems can orchestrate robust collaborative governance mechanisms to design long-term smart and sustainable transitions, by reinforcing both the exploitation of the drivers and the minimization of the barriers. The focus was on the three key drivers, as identified above; nonetheless, it is worth mentioning again that many more combinations exist.

## 5. Conclusions

The evolution of innovation ecosystems models (i.e., from Triple to Quintuple Helix) yields for the parallel to the evolution of smart urban governance approaches. The state-of-the-art review confirms that research is already being conducted toward this direction, yet the complexity imposed by the emerging innovation ecosystems demands the perpetual reinforcement of governance mechanisms. At the same time, the conceptualization of most appropriate, dynamic and customized frameworks to the smart city transitions or even to the specific smart project to be envisaged.

The selected case studies revealed various findings. To begin with, the Triple Helix case revealed that depending on the nature and the scope of the project, public participation might not be necessary for securing the success of the project. Nevertheless, we highlight that public participation is recommended both for its dynamic for innovation and value creation (e.g., via open participation and open innovation schemes), as well as for the general elevation to a knowledge society (and knowledge economy and knowledge democracy) with 'smart' citizens. In doing so, however, it is crucial to engage the public as early as the planning stage of the project to raise their awareness and to ensure their continued involvement. This must be accomplished through consistent and transparent communication between citizens and all stakeholders involved to build trust and credibility. Failure to do so may negatively impact the realization of the project, as demonstrated in the Flottsund bridge renovation project. Finally, the Quintuple Helix case study confirmed the maturity of the whole framework and suggested that perhaps of the time and the holistic approach it offers to sustainable development, smart city governances are learning from past literature and empirical mistakes and adapting to new global conditions and addressing urban challenges efficiently and effectively. Nonetheless, as a future work, it is highly recommended to undergo a more extensive and exhausting literature review on stories of success and failure per Helix and create a roadmap of the empirical evolution of the Helix concept. We furthermore suggest that the smart city governance 'ecosystems' should not only be examined internally but also externally with their interoperability with other urban cases as the knowledge, innovation, and creativity are only accelerated by sharing and interacting within complex networks that strive for the same long-term goals (e.g., open innovation 2.0).

The power of innovation ecosystems was complementarily elaborated from the perspective that they allow the emergence of multiple potential solutions where strengths, weaknesses, opportunities, and threats of smart governance mechanisms can be orchestrated for ameliorating the design and implementation of smart city projects toward sustainable development. Even though the authors chose to focus on the so-called key drivers for smart city development for the purposes of this review, it is suggested that the multiple relationships between the different areas of the SWOT analysis should be further explored and that the case studies analysis will become exhaustively investigated in more cases as well in a benchmarking method. The limitations on data and previous lessons-learnt have been the main restrictions of this study, as well as the complexity of the

transferability of these models to real cases. Further and detailed approaches are requested to better understand and encode the powerful impacts on 'ecosystems' collaboration' to achieve smartness in our future cities.

The smart city field may be studied at each level and scale from the regional to the more global emerging and rising demands of the cities' populations. In this manuscript, we emphasized the role of the organization, while given its reviewing character, an analytical process of the n-Helix models with a critical angle is identified as a powerful lever for innovation. Keywords, such as 'governance', 'ecosystem', or 'collaboration', enabled us to display the role of 'smart ecosystems' for solutions against challenging phenomena, such as the rapid urbanization or the climate change. The study, through the literature review and the benchmarking process of selected and successful European projects of the models' applied, revealed the lack of exhaustive analyses for the methodological investigation, identification, and adoption of the most appropriate governance model and collaborative approaches per project and collaborative approaches and create modular frameworks; nonetheless, more evolutionary frames based on the development of synergies of stakeholders are required to prioritize their importance in the smart city planning.

**Author Contributions:** All the authors contributed to the paper writing. P.R.-F., M.A.P. and R.T. worked mainly at the data collection and the representation of ecosystem models and the presented case studies, while S.K., M.P.-E., D.T. and R.T. reviewed the general context of the paper. All authors have read and agreed to the published version of the manuscript.

**Funding:** This research received no external funding.

**Institutional Review Board Statement:** The study was conducted according to the guidelines of the Declaration of Helsinki, and approved by the Institutional Review Board (or Ethics Committee).

**Informed Consent Statement:** Informed consent was obtained from all subjects involved in the study.

**Data Availability Statement:** Not applicable.

**Acknowledgments:** This research was supported by the Erasmus Mundus Joint Master Program SMACCs (Smart Cities and Communities) and the University of Mons.

**Conflicts of Interest:** The authors declare no conflict of interest.

## Nomenclature

| Abbreviation | Definition |
| --- | --- |
| ICT | Information and Communication Technologies |
| QoL | Quality of Life |
| SSC | Sustainable Smart City |
| IT | Information Technology |
| SDG | Sustainable Development Goal |
| NGO | Nongovernmental Organization |
| SME | Small and Medium Enterprises |
| SWOT | Strengths, Weaknesses, Opportunities, and Threats |

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
