# Peer review of "Unveiling the Evolution of Innovation Ecosystems: An Analysis of Triple, Quadruple, and Quintuple Helix Model Innovation Systems in European Case Studies"

_sustainability, doi:10.3390/su13147582_

Round 1
Reviewer 1 Report
This manuscript presents an exhaustive analysis on the importance of the n-Helix models along with a benchmarking critical approach through selected European case-studies through the literature review and case studies. It is well structured and written. The results are clearly explained and presented in an appropriate format. However, some minor improvements should be addressed:
1. The figures should be well modified for better clearness.
2. Line 603: Lack of a full stop, ".", at the end of the sentence.
3. In addition, the style of some references may be revised.
Reviewer 2 Report
Authors present a research about the n-Helix models (triple, quadruple and quintuple) and the analysis of three different case-studies. The core-idea of the paper is quite interesting and very relevant in the nowadays context but, in my opinion, it is not very clear the methodology and the research framework authors use in their dissertation.
The main issue is that it is difficult to understand if authors identify the 3 case studies as simple examples of n-Helix model or if they say that they have been conceived, designed and developed starting from the concepts expressed by n-Helix model itself. Moreover, does the n-Helix model describe or influence the project development process? In the final step of their research, authors make a SWOT analysis that I do not see directly linked to the n-helix model: the projects can be analyzed in this way at first. The discussion section is very well-done and highlight problems and driving forces in the smart cities and ecosystems developing process focusing on the governance and public (society) point of view.
Minor issues:
- Do authors create Figure 3 or take it from other researches? please explain;
- In Figure 4 (from reference 26 but it is not an image directly from that paper) the factor "other policy domains" appear twice; please explain.
Reviewer 3 Report
Dear Authors,
There are few changes that would needed to be performed on your manuscript to increase its value. Please find my comments below:
The quality of the figures included in the manuscript should be improved, as they seem to appear pixelated. For instance, the words in Figure 4 cannot be clearly read. Please try and amend this aspect for all figures.
Page 10: "The Quadruple Helix model, from the explanatory potential ...[53]". The message is not clear, please rephrase.
Indicate the source for each figure. If it is created by you, specify this.
Add the limitations of the study and future research directions in the section dealing with concluding remarks.
Please format the references in line with the journal requirements.
Round 2
Reviewer 2 Report
I want to thank authors for the precise and exhaustive answers to my doubts and suggestions. They made a huge effort to improve the paper quality, clarity and readability.
Although I do not believe that "the n-Helix model influences the project development process" consciously, I am quite pleased with the paper's update version.
In my opinion it is now ready to be published.